# Biomarker–Sleep Correlations in PTSD: Glutamine, Glutathione, Caspase-1, and BDNF Levels Assessed Using the Pittsburgh Sleep Quality Index Addendum

**DOI:** 10.3390/cimb47100814

**Published:** 2025-10-02

**Authors:** Anna Dorota Grzesińska

**Affiliations:** Collegium Medicum, Jan Dlugosz University in Częstochowa, Waszyngtona 4/8 Street, 42-200 Częstochowa, Poland; a.grzesinska@ujd.edu.pl

**Keywords:** BDNF, caspase-1, glutamine, glutathione, post-traumatic stress disorder

## Abstract

Emerging evidence highlights oxidative stress and its biomarkers as potential factors in the onset and maintenance of Post-Traumatic Stress Disorder (PTSD) and co-occurring sleep disturbances. The study concerns the profile of biomarkers including glutamine, glutathione (GSH), caspase-1 and Brain-Derived Neurotrophic Factor (BDNF) levels in three groups (PTSD with a current diagnosis lasting ≤ 5 years, PTSD with a current diagnosis lasting > 5 years, and no PTSD), classified into two age groups. In addition, sleep disturbances were analyzed using the Pittsburgh Sleep Quality Index Addendum (PSQI-A). The study revealed mutual correlations between the examined biomarkers, which may confirm a coordinated antioxidant response. Furthermore, a relationship was observed between biomarkers and PSQI-A; trauma-related domains (e.g., Trauma Nightmares with Terror Episodes) were more pronounced in the case of PTSD ≤ 5 years, while PTSD > 5 years emphasized trauma-unrelated anxiety. The study results suggest that individuals with PTSD exhibit increased sensitivity to trauma, which may manifest through immune system activation and sleep disturbances. Patients with a longer history of PTSD and co-occurring dysfunctions require a personalized approach to trauma treatment and prevention of recurrence.

## 1. Introduction

Stress is a form of burdening experience for the body, involving a disruption of homeostatic balance, and its accumulation may lead to allostatic load disturbances. When allostatic processes are excessively activated by chronic stress, the body becomes unable to effectively manage health regulation and loses its regenerative capacity, which can result in an allostatic load and the development of diseases [1,2]. Research shows that stress affects humans in a bidirectional manner. Acute stress can stimulate the immune system, whereas chronic stress may significantly weaken it, cause disturbances in neurohormonal regulation, and promote the development of neuroinflammation [3,4,5,6]. In Post-Traumatic Stress Disorder (PTSD), stress is no longer a one-time reaction to a threat but becomes a long-lasting, often debilitating disorder that occurs after experiencing a traumatic event. It is a mental disorder triggered by trauma, manifesting through intrusive thoughts, recurring nightmares, and avoidance of certain situations. It is accompanied by deep emotional and psychological distress, often significantly impacting daily life. Notably, PTSD may lead to molecular changes in the body, including disruptions in redox balance, cellular damage, and increased inflammatory symptoms [7,8].

One of the key components involved in the body’s antioxidant defense is glutathione. This compound is a tripeptide composed of a γ-peptide bond between cysteine and the carboxyl group of the glutamate side chain. The carboxyl group of the cysteine residue is bonded to glycine via a standard peptide bond. Glutathione exists in the body in both reduced (GSH) thiol and the oxidized (GSSG) disulfide forms [9,10]. The ratio of reduced to oxidized glutathione in cells determines the intensity of oxidative stress. An increased GSSG-to-GSH ratio indicates heightened oxidative stress. Antioxidants like glutathione play a crucial role in neutralizing reactive oxygen species (ROS), thus preserving redox homeostasis and protecting cells from oxidative damage. As the most abundant intracellular antioxidant in the central nervous system, glutathione is a key indicator of oxidative stress. During oxidative challenges, two molecules of reduced glutathione each donate an electron, forming oxidized glutathione. This oxidized form can either be recycled back to GSH by the enzyme glutathione reductase (GR) or expelled from the cell through an ATP-dependent transporter in the plasma membrane to prevent toxic buildup. Notably, reduced levels of GSH have been linked to the pathophysiology of PTSD [11].

It is known that glutamine is a key precursor for glutathione synthesis. Glutamine is converted into glutamate by the enzyme glutaminase, after which glutamate, together with cysteine and glycine, is used for glutathione production. This process plays a crucial role in maintaining antioxidant levels and protecting cells from oxidative stress [12]. Glutamine is also the main precursor of glutamate, a key excitatory neurotransmitter. In the brain, glutamate is the most widely used excitatory neurotransmitter in all signal transmission processes. During synaptic transmission, glutamate released from presynaptic neurons via the glutamate–glutamine cycle activates postsynaptic glutamate receptors, thereby mediating excitatory signaling. This cycle also effectively removes glutamate from the synaptic cleft and maintains a relatively low concentration, preventing excitotoxic damage caused by excessive Ca^2+^ influx into postsynaptic neurons due to receptor overactivation [13]. Disruptions in synapse formation, abnormalities in glutamatergic signaling, and altered neural circuitry development can contribute to the persistence of PTSD symptoms [14,15].

Brain-derived neurotrophic factor (BDNF) is a key protein that supports neuronal survival, synaptic plasticity, and learning processes. Evidence shows that individuals with PTSD often present reduced levels of BDNF in both blood and brain regions such as the hippocampus and prefrontal cortex [16,17,18]. Low BDNF may impair the brain’s ability to regulate fear memories, weaken the extinction of traumatic associations, and disrupt memory consolidation, which can contribute to persistent nightmares and poor sleep quality [19]. In addition, reduced BDNF can interact with oxidative stress and neuroinflammatory processes, leading to further dysregulation of neural circuits involved in stress and sleep regulation [20]. These findings suggest that BDNF is not only a marker of neurobiological changes in PTSD but also a potential target for therapy. Treatments that increase BDNF activity—such as physical exercise, certain antidepressants, or neuromodulation—may help restore brain plasticity and improve sleep in patients with PTSD.

Additionally, the caspase-1/interleukin-1β converting enzyme (ICE) is involved in the neuroinflammatory process. Caspase-1 is located in various intracellular compartments, such as the cytoplasm, vesicles, and even the nucleus, depending on the cell’s activation status [21]. It is activated by inflammasome complexes, which are intracellular sensors of danger signals. Caspase-1 activation leads to the maturation of cytokines interleukin-1β (IL-1β) and interleukin-18 (IL-18) and triggers pyroptosis, a form of programmed cell death. It plays a key role in initiating the inflammatory response in the development of cellular immunity. Caspase-1 acts as a protease responsible for converting the inactive IL-1β precursor into its active, secreted form [22]. Numerous scientific publications highlight the importance of this enzyme as a key mediator of inflammatory processes, including neuroinflammation in PTSD [23,24].

Sleep disturbance, a hallmark symptom of PTSD, may serve as a key pathway linking the disorder to elevated neuroinflammatory serum biomarkers. Numerous studies have shown that both subjective complaints and objective measures of disrupted sleep—including insomnia that occurs before or after trauma—are predictive of subsequent PTSD development. Sleep plays a vital role in brain health, supporting essential cellular functions such as detoxification and repair. Its restorative capacity stems from reduced neural activity during sleep, including decreased glucose metabolism and oxidative processes, which permits antioxidant and anti-inflammatory mechanisms to address the metabolic demands accumulated during wakefulness [25,26].

Glutamine, glutathione, caspase-1, and BDNF are among the most frequently studied biomarkers in PTSD, strongly associated with oxidative stress, inflammation, and neurotrophic processes, although not all studies have fully confirmed these associations [7,12,13,14,16,19,20,21].

In the context of PTSD, a thorough understanding of its complex neurobiology and the development of effective pharmacotherapies is essential. There is growing interest in targeting neurokinin-1 receptors, melanocortin-2 receptors, oxytocin receptors, and components of the endocannabinoid system as potential therapeutic strategies. The authors stress the need for further clinical studies to evaluate the safety and efficacy of emerging pharmacological treatments. They also emphasize the importance of integrating pharmacotherapy with psychological interventions, such as cognitive–behavioral therapy, to enhance treatment outcomes [27].

This study aimed to:Characterize the demographic and addiction-related profiles of male individuals with a current diagnosis of PTSD lasting ≤ 5 years (PTSD ≤ 5 y) and >5 years (PTSD > 5 y) compared to a control group with no diagnosis of PTSD, focusing on differences in age, occupational exposure, body mass index (BMI), and addiction behaviors.Investigate the biomarker profiles of glutamine, glutathione (GSH), caspase-1 expression, and BDNF across the three groups (PTSD ≤ 5 y, PTSD > 5 y, and No PTSD), stratified by age (18–50 years, with subgroups 18–35 and 36–50 years), to assess variations associated with PTSD chronicity and age.Evaluate sleep disturbance patterns using the Pittsburgh Sleep Quality Index Addendum (PSQI-A) across the three groups, stratified by age, to determine the impact of PTSD chronicity and age on specific sleep domains and overall sleep quality.Examine the correlations between biomarkers (glutamine, glutathione, caspase-1, BDNF) and PSQI-A domains within each group to elucidate the relationship between biomarker dysregulation and sleep disturbances, and to explore how these relationships vary with PTSD chronicity.Identify major differences in biomarker–PSQI-A correlations among the groups to understand the evolving biological and clinical profiles of PTSD over time, providing insights into potential pathophysiological mechanisms and therapeutic targets.

## 2. Material and Methods

### 2.1. Characteristics of the Participants

The study included 92 adult male participants divided into three groups based on PTSD duration: individuals with a current diagnosis of PTSD lasting ≤ 5 years (PTSD ≤ 5 y, n = 33), those with a current diagnosis of PTSD lasting > 5 years (PTSD > 5 y, n = 31), and a control group with no diagnosis of PTSD (No PTSD, n = 28). All participants were recruited from occupational environments characterized by a high risk of trauma exposure, and recruitment was conducted through publicly available media announcements. This ensured relative homogeneity in trauma-related risk factors. Inclusion criteria were: age 18–50, current or past exposure to traumatic events, and employment in high-risk occupations. Exclusion criteria included chronic somatic or psychiatric illnesses (except for compulsive behaviors), medication use, and the presence of confounding factors that could affect biomarker measurements. All participants were physically healthy, did not take any medication, and reported no chronic diseases. Written informed consent was obtained from all participants prior to enrollment.

The diagnosis of PTSD was established according to DSM-5 criteria using the Clinician-Administered PTSD Scale for DSM-5 (CAPS-5).

Addiction-related and compulsive behaviors were assessed through structured clinical interviews based on DSM-5 criteria for impulse-control and non-substance-related addictive disorders. The interviews included items on compulsive eating, intrusive trauma-related visualization, excessive exercise, and cyberaddiction.

### 2.2. Pittsburgh Sleep Quality Index Addendum

Sleep quality over the past month was assessed using the Pittsburgh Sleep Quality Index (PSQI), a validated instrument designed to capture trauma-related sleep disturbances, including nightmares and nocturnal anxiety symptoms [25]. The scale measures seven domains of sleep disturbance: hot flashes, general nervousness, trauma-related memories or nightmares, non-trauma-related anxiety or panic, distressing non-trauma-related dreams, episodes of terror, and acting out dreams [25].

These components are summed to generate a global score ranging from 0 (indicative of good sleep quality) to 21 (poor sleep quality). A total score of 5 or less reflects good sleep quality, while scores above 5 indicate poor sleep. The PSQI demonstrates strong psychometric properties, with a sensitivity of 89.6% and specificity of 86.5%.

To specifically assess sleep disturbances associated with PTSD, the PSQI-A was used. The PSQI-A consists of seven items that capture disruptive nocturnal behaviors frequently reported by individuals with PTSD, such as hot flashes, trauma-related nightmares or memories, and episodes of sleep-related terror. Each item is rated on a 4-point scale from 0 (not during the past month) to 3 (three or more times per week), yielding a total score ranging from 0 (no disturbance) to 21 (severe disturbance). A score of 4 or higher is considered indicative of probable PTSD, based on initial validation findings. The PSQI-A has demonstrated excellent diagnostic accuracy, with a sensitivity of 94% and specificity of 82%.

### 2.3. Clinician-Administered PTSD Scale for DSM-5 (CAPS-5)

To assess the severity of PTSD symptoms, we used the Clinician-Administered PTSD Scale for DSM-5 (CAPS-5), administered by the study authors—trained clinical psychologists—in accordance with the standardized administration and scoring protocol. CAPS-5 is widely regarded as the gold standard for PTSD diagnosis and symptom assessment. It includes 20 items corresponding to the DSM-5 diagnostic criteria, with each symptom rated for both frequency and intensity on a 5-point scale (0–4). The total symptom severity score ranges from 0 to 80.

Interpretation of total scores followed the classification proposed by Weathers et al. [24], where

11–22: Mild PTSD symptoms;23–34: Moderate PTSD symptoms;35–60: Moderately severe PTSD symptoms;>60: Severe PTSD symptoms.

### 2.4. Blood Sampling

At the time of enrollment, serum concentrations of glutamine, glutathione, caspase-1, and BDNF were measured once per participant using a commercially available enzyme-linked immunosorbent assay (ELISA), conducted according to the manufacturer’s protocol. This sandwich-type immunoassay utilizes monoclonal anti-human antibodies for target detection. Blood samples were obtained via standard venipuncture procedures, and serum was isolated through centrifugation. The resulting aliquots were stored at −80 °C to preserve sample integrity and prevent cytokine degradation. Prior to analysis, serum samples were thawed, gently vortexed, and diluted at a ratio of 1:3 using the supplied Dilution Buffer to ensure optimal concentrations within the assay’s dynamic detection range.

### 2.5. Biomarker Analysis Procedure

Each assay included standards, blank controls, and appropriately diluted serum samples, which were pipetted into microtiter plates pre-coated with specific capture antibodies. Samples were incubated at room temperature for 60 min with continuous agitation (300 rpm). Following an initial wash, a biotin-labeled detection antibody was added, and plates were incubated for an additional 60 min. After a second washing step, Streptavidin conjugated to horseradish peroxidase (HRP) was introduced and incubated for 30 min. Subsequent to a final wash, 100 µL of substrate solution was added to each well and incubated for 10 min at room temperature. The enzymatic reaction was halted by the addition of an acidic stop solution. Absorbance was then measured at 450 nm using a microplate reader.

### 2.6. Assay Sensitivity and Standardization

A calibration curve was established using recombinant standards for each biomarker. The catalog numbers of reagents, assay detection ranges, and sensitivity thresholds are listed below.

Glutamine, Glutathione, Caspase-1, and BDNF.

Glutamine: Assay range 0.15–40 ng/mL; sensitivity 0.108 ng/mL; (SunRedBio, Shanghai, China); Catalogue No 201-12-8432.

Glutathione: Assay range 0.2 Umol/L–30 Umol/L; sensitivity 0.125 Umol/L; (SunRedBio, Shanghai, China); Catalogue No 201-12-1463.

Caspase-1: Assay range 0.15–32 ng/mL; sensitivity 0.108 ng/mL; (SunRedBio, Shanghai, China); Catalogue No 201-12-0791.

BDNF: Assay range 0.1–10 ng/mL; sensitivity 0.05 ng/mL; (SunRedBio, Shanghai, China); Catalogue No 201-12-1303.

### 2.7. Statistical Analysis

A significance level of α = 0.05 was used for all statistical tests, with *p*-values < 0.05 considered statistically significant. Descriptive statistics were reported as medians with interquartile ranges (IQR, Q1–Q3) for continuous variables due to non-normal distribution, as assessed by the Shapiro–Wilk test (*p* < 0.05 for most variables). For PSQI-A scores, consisting of ordinal values with a limited range (typically 0–3 per domain), means with standard deviations (SD) were reported to enhance readability. Categorical variables, if present, would be reported as frequencies and percentages.

Group differences in demographic and addiction-related data across the three groups (PTSD ≤ 5 y, PTSD > 5 y, and No PTSD) were assessed using the Kruskal–Wallis rank sum test. Post hoc pairwise comparisons employed Dunn’s test with Bonferroni correction, denoted by compact letter display (e.g., a, b, c), where shared letters indicate no significant difference (adjusted *p* > 0.05). Age subgroup differences (18–35 vs. 36–50 years) used the Wilcoxon rank-sum test with Holm–Bonferroni correction across groups.

Group differences in biomarker data (glutamine, glutathione, caspase-1, and BDNF) were assessed using Quade’s rank-based ANCOVA to adjust for BMI confounding and non-normality. Post hoc pairwise comparisons applied Quade’s test on subsets with Holm–Bonferroni correction, using compact letter display for significance.

Group differences in PSQI-A scores were assessed using ANCOVA to adjust for BMI confounding, with Welch’s ANOVA accounting for unequal variances. Post hoc comparisons used the Games–Howell test with Holm–Bonferroni correction, denoted by compact letter display. Age subgroup differences in PSQI-A scores employed Welch’s t-test with Holm–Bonferroni correction.

For correlation analyses between biomarkers and PSQI-A domains, Spearman’s rank correlation coefficient (ρ) was used due to non-normality, with two-tailed *p*-values reported to assess significance. The correlation analysis was conducted for exploratory purposes; therefore, no corrections for multiple comparisons were applied, as such adjustments in exploratory contexts may inflate Type II error rates and obscure potentially meaningful patterns, particularly when examining interdependent biomarkers in novel PTSD subgroups [28]. This approach prioritizes sensitivity to detect preliminary associations for hypothesis generation, with results interpreted cautiously to inform future confirmatory research.

Missing data were minimal (e.g., N = 90–91 for some addiction-related variables), and analyses were conducted on available data without imputation. Sample size (N = 92) was determined based on prior studies of PTSD biomarker profiles, ensuring sufficient power to detect moderate effect sizes (Cohen’s d = 0.5) at α = 0.05 and 80% power.

### 2.8. Characteristics of the Statistical Tool

Analyses were conducted using the R Statistical language, version 4.3.3 [29] on Windows 11 Pro 64 bit (build 26100), using the packages Hmisc, version 5.2.0 [30]; ggpubr version 0.6.0 [31]; report version 0.5.8, [32]; RColorBrewer (version 1.1.3 [33]; gtsummary version 1.7.2, [34]; corrplot version 0.94 [35], ggplot2 version 3.5.0 [36]; stringr version 1.5.1, [37]; dplyr version 1.1.4 [38]; and tidyr version 1.3.1 [39].

## 3. Results

### 3.1. Characteristics of Demographic and Addiction-Related Profiles in PTSD

The study sample comprised 92 male participants, categorized into three groups based on PTSD status: PTSD with a current diagnosis lasting ≤ 5 years (PTSD ≤ 5 y, n = 33), PTSD with a current diagnosis lasting > 5 years (PTSD > 5 y, n = 31), and a control group with no diagnosis of PTSD (No PTSD, n = 28). Demographic and addiction-related characteristics are summarized in Table 1.

#### 3.1.1. Demographic Profile

Demographically, the groups were comparable in both age (median: 34–36 years) and duration of occupational exposure (median: ~10 years), suggesting that observed differences in clinical or biological measures are unlikely to be confounded by chronological age or work-related factors.

With respect to addiction-related behaviors, individuals with PTSD—regardless of disorder duration—showed a significantly higher prevalence and longer duration of behavioral addictions compared to controls. These included compulsive behaviors, disordered eating, pathological visualization (e.g., intrusive and repetitive trauma-related imagery), and exercise addiction. Median durations for these behaviors ranged from 1.0 to 4.0 years in the PTSD groups, while they were negligible or absent in the control group. Behavioral patterns were assessed through structured clinical interviews based on DSM-5 criteria for impulse-control and non-substance-related addictive disorders. Statistical analyses revealed significant group differences across all addiction domains except for cyberaddictions.

The chronic PTSD group (current PTSD diagnosis lasting > 5 years; PTSD > 5 y) reported the longest durations of compulsive behavioral patterns, suggesting that prolonged PTSD may increase vulnerability to persistent maladaptive coping strategies. However, the early PTSD group (current PTSD diagnosis lasting ≤ 5 years; PTSD ≤ 5 y) already displayed addiction behavior profiles comparable to the chronic group, indicating that such tendencies emerge soon after trauma exposure and tend to stabilize over time.

#### 3.1.2. Addiction-Related Profile

Participants with a history of PTSD exhibited a markedly higher burden of addiction-related behaviors compared to controls, with the duration and type of addiction varying by PTSD chronicity. Both PTSD groups showed elevated durations of compulsive behavioral, eating-related, pathological visualization, and exercise compulsion addictions (medians 1.0–4.0 years) relative to the No PTSD group (median 0.0 years), with statistically significant differences in all but cyberaddictions. The PTSD > 5 y group (current diagnosis lasting more than 5 years) stood out with the longest duration of compulsive behavioral addictions (4.0 years), demonstrating that chronic PTSD may amplify vulnerability to persistent, maladaptive coping mechanisms over time. In contrast, the PTSD ≤ 5 y group (current diagnosis lasting 5 years or less) aligned closely with the > 5 y group across most addiction measures (sharing B designations), indicating that addiction tendencies emerge early after PTSD onset and may stabilize rather than diminish or escalate beyond 5 years.

#### 3.1.3. CAPS-5 Score

The median CAPS-5 total score for the entire sample was 57.5, with IQR 10.5–67.0. Among participants with PTSD ≤ 5 y (current diagnosis lasting 5 years or less) (N = 33), the median score reached 68.0 (IQR: 66.0–72.0), denoting severe symptom intensity with minimal intragroup variation. In contrast, those with PTSD > 5 y (current diagnosis lasting more than 5 years) (N = 31) exhibited a median score of 56.0 (IQR: 54.0–59.0), indicative of moderate to severe symptoms accompanied by modestly increased variability. The control group without posttraumatic stress disorder (N = 28) displayed a median score of 7.0 (IQR: 4.5–9.0), reflecting negligible symptoms as expected. Intergroup disparities were profoundly significant (*p* < 0.001), delineating a clear severity hierarchy: PTSD ≤ 5 y (current diagnosis lasting 5 years or less) than PTSD > 5 y (current diagnosis lasting more than 5 years), which in turn exceeded controls, and all pairwise contrasts proving distinct. This pronounced gradient underscores the rationale for stratifying patients by posttraumatic stress disorder duration, as shorter exposure intervals associate with heightened symptom burden, whereas longer durations evince partial attenuation yet sustained clinical relevance, thereby justifying differentiated categorization in research frameworks.

### 3.2. Biomarker Profiles in PTSD: Age-Stratified Analysis

Based on the results in Table 2, biomarker levels, controlled by BMI, differed significantly across the three groups (*p* < 0.001) within each age stratum. Glutamine and glutathione (GSH) levels were lowest in the PTSD ≤ 5 y group (e.g., glutamine: 4.55 ng/mL [IQR: 3.46–5.40]; GSH: 1.67 nmol/mL [IQR: 1.27–2.24] for 18–50 years), intermediate in the PTSD > 5 y group (glutamine: 10.72 ng/mL [IQR: 7.70–21.07]; GSH: 5.52 nmol/mL [IQR: 3.33–7.89]), and highest in the No PTSD group (glutamine: 42.52 ng/mL [IQR: 31.18–45.68]; GSH: 13.69 nmol/mL [IQR: 11.19–19.21]), indicating a progressive normalization with time since PTSD diagnosis, with all pairwise differences significant (adjusted *p* < 0.001). Conversely, caspase-1 levels were highest in the PTSD ≤ 5 y group (25.70 ng/mL [IQR: 19.72–28.57]), intermediate in the PTSD > 5 y group (6.09 ng/mL [IQR: 3.53–17.23]), and lowest in the No PTSD group (2.77 ng/mL [IQR: 2.40–3.29]), revealing greater inflammation in recent PTSD cases; all pairwise differences were significant (adjusted *p* ≤ 0.030) except in the 36–50 years subgroup, where the difference between the PTSD ≤ 5 y (23.64 ng/mL [IQR: 19.60–28.57]) and PTSD > 5 y (9.37 ng/mL [IQR: 3.40–21.62]) groups was non-significant (adjusted *p* = 0.051). BDNF levels were lowest in the PTSD > 5 y group (2.52 ng/mL [IQR: 1.88–2.73]), intermediate in the PTSD ≤ 5 y group (3.33 ng/mL [IQR: 2.75–3.56]), and highest in the No PTSD group (5.61 ng/mL [IQR: 5.37–6.10]), consistent with impaired neurotrophic support in PTSD.

Age stratification revealed subtle variations. For glutathione (GSH), levels were significantly higher in the 18–35 years subgroup compared to the 36–50 years subgroup in the PTSD ≤ 5 y group (2.15 nmol/mL vs. 1.38 nmol/mL, adjusted *p* = 0.021) and in the No PTSD group (17.55 nmol/mL vs. 12.61 nmol/mL, adjusted *p* = 0.042), while in the PTSD > 5 y group, levels were higher in the older subgroup but the difference was not statistically significant (4.58 nmol/mL vs. 6.32 nmol/mL, adjusted *p* = 0.141). For BDNF, levels were higher in the 36–50 years subgroup compared to the 18–35 years subgroup in the No PTSD group (5.76 ng/mL vs. 5.56 ng/mL, adjusted *p* = 0.672) and in the PTSD ≤ 5 y group (3.46 ng/mL vs. 2.96 ng/mL, adjusted *p* = 0.130); however, the PTSD > 5 y group showed lower levels in the 36–50 years subgroup (2.43 ng/mL vs. 2.61 ng/mL, adjusted *p* = 0.672), but none of these differences were statistically significant after adjustment for multiple comparisons. Glutamine and caspase-1 showed consistent patterns across age groups, with no significant differences between age subgroups within any group (all adjusted *p* > 0.05). Overall, the age-group factor significantly influenced glutathione levels in the PTSD ≤ 5 y and No PTSD groups but did not substantially affect the other biomarkers across the groups.

Biomarker levels differed significantly across the three groups (*p* < 0.001) within each age stratum. Glutamine and glutathione were lowest in individuals with a current diagnosis of PTSD lasting ≤ 5 years (PTSD ≤ 5 y, n = 33; 4.55 ng/mL; 1.67 nmol/mL), intermediate in those with a current diagnosis of PTSD lasting > 5 years (PTSD > 5 y, n = 31; 10.72 ng/mL; 5.52 nmol/mL), and highest in the control group with no diagnosis of PTSD (No PTSD, n = 28; 42.52 ng/mL; 13.69 nmol/mL). Caspase-1 was highest in PTSD ≤ 5 y (25.70 ng/mL), intermediate in PTSD > 5 y (6.09 ng/mL), and lowest in controls (2.77 ng/mL). BDNF showed the lowest values in PTSD > 5 y (2.52 ng/mL), intermediate in PTSD ≤ 5 y (3.33 ng/mL), and the highest in controls (5.61 ng/mL). Age stratification revealed only minor differences in medians, without altering the overall group patterns (Figure 1).

### 3.3. PSQI-A Scores Across Control and PTSD Groups by Age Distribution

The current analysis examines PSQI-A scores across three groups—individuals with a current PTSD diagnosis lasting ≤ 5 years (PTSD ≤ 5 y), those with a current PTSD diagnosis lasting > 5 years (PTSD > 5 y), and a control group with no diagnosis of PTSD (No PTSD)—stratified by age (18–50 years, with subgroups 18–35 and 36–50 years). The analysis aims to elucidate differences in sleep disturbance patterns and their potential relationship with age and PTSD chronicity, providing insights into the long-term impact of PTSD on sleep quality.

According to results in Table 3, both PTSD groups consistently exhibited higher mean scores (2.00–2.94) across all domains compared to the No PTSD group (0.00–1.29), reflecting severe sleep disturbances in individuals with PTSD (e.g., PTSD ≤ 5 y vs. No PTSD in Hot Flashes, 18–50 years: adjusted *p* < 0.001; PTSD > 5 y vs. No PTSD: adjusted *p* < 0.001). The No PTSD group showed minimal disruption, with mean scores often at 0.00, particularly in trauma-related domains like Memories/Nightmares of Trauma (Trauma NM), underscoring the specificity of these symptoms to PTSD (e.g., Trauma NM, 18–50 years: PTSD ≤ 5 y vs. No PTSD, adjusted *p* < 0.001; PTSD > 5 y vs. No PTSD, adjusted *p* < 0.001).

For most domains, including General Nervousness, Trauma NM, Anxiety/Panic Not Related to Trauma (Non-Trauma Anx), and Distressing Dreams Not Related to Trauma (Non-Trauma Dr), the PTSD ≤ 5 y and PTSD > 5 y groups displayed comparable severity (means of 2.20–2.94), with no significant difference between them (e.g., General Nervousness, 18–50 years: PTSD ≤ 5 y vs. PTSD > 5 y, adjusted *p* = 0.836; Trauma NM, 18–50 years: adjusted *p* = 0.609). Exceptions were noted in the Hot Flashes domain, where the PTSD ≤ 5 y group had a significantly higher mean score (2.55) compared to PTSD > 5 y (2.13) in the 18–50 years group (adjusted *p* = 0.046), and specifically in the 18–35 years subgroup (2.75 vs. 1.60, adjusted *p* < 0.001), reflecting attenuation of this symptom over time in younger individuals; however, in the 36–50 years subgroup, both PTSD groups reported comparable means (2.35 vs. 2.63, adjusted *p* = 0.232), indicating persistence in older age. Furthermore, in the Episodes of Terror domain, the PTSD ≤ 5 y group showed a significantly higher mean score than PTSD > 5 y in the 36–50 years subgroup (2.47 vs. 1.94, adjusted *p* = 0.014), though comparable in other strata (adjusted *p* = 0.383). Similarly, in Acting Out Dreams, the PTSD ≤ 5 y group had significantly higher means than PTSD > 5 y in the 18–50 years (2.48 vs. 2.00, adjusted *p* = 0.002) and 36–50 years (2.47 vs. 1.94, adjusted *p* = 0.014) subgroups, but comparable in 18–35 years (adjusted *p* = 0.066).

Total PSQI-A scores further confirmed these patterns, with PTSD groups scoring 15.93–18.31 across age subgroups compared to 2.64 in the No PTSD group (e.g., 18–50 years: PTSD ≤ 5 y vs. No PTSD, adjusted *p* < 0.001; PTSD > 5 y vs. No PTSD, adjusted *p* < 0.001). The PTSD ≤ 5 y group had a significantly higher total score than PTSD > 5 y in the 18–50 years (18.00 vs. 16.68, adjusted *p* < 0.017) and 18–35 years (18.31 vs. 15.93, adjusted *p* < 0.002) subgroups, but comparable in 36–50 years (17.71 vs. 17.38, adjusted *p* = 0.740).

Age-related differences within groups were subtle; the PTSD > 5 y group showed a slightly lower mean total score (15.93) in the 18–35 years subgroup compared to 17.38 in the 36–50 years subgroup, but this difference was not statistically significant (*p* = 0.340), while the PTSD ≤ 5 y group showed stable total scores (18.31–17.71, *p* = 0.688).

Within the No PTSD group, total scores were identical in mean (2.64) across age subgroups (*p* = 1.000), though with greater variability in the 36–50 years subgroup (SD = 4.50) compared to the 18–35 years subgroup (SD = 1.91). Examining specific domains across age subgroups, the 18–35 years subgroup showed variability in Hot Flashes (PTSD ≤ 5 y vs. PTSD > 5 y, significant difference as indicated by different superscripts), revealing that younger individuals with recent PTSD may experience more pronounced physical symptoms in some cases. For General Nervousness, older individuals reported slightly greater scores irrespective of PTSD duration (PTSD ≤ 5 y: 2.53 vs. 2.38 in younger, *p* = 0.986; PTSD > 5 y: 2.63 vs. 2.20 in younger, *p* = 0.231), but these age differences were not statistically significant. In contrast, the PTSD ≤ 5 y group exhibited significantly higher scores in Distressing Dreams Not Related to Trauma in the 36–50 years subgroup (2.65 vs. 2.19, *p* = 0.018), while the PTSD > 5 y group showed a non-significant trend toward higher scores (2.75 vs. 2.33, *p* = 0.075). Additionally, in the PTSD > 5 y group, Hot Flashes scores were significantly higher in the 36–50 years subgroup (2.63 vs. 1.60, *p* = 0.004), while Episodes of Terror showed a non-significant decrease with age (1.94 vs. 2.33, *p* = 0.150).

Domains like Trauma Nightmares (Trauma NM) showed consistent severity across age subgroups within PTSD groups (e.g., Trauma NM, PTSD ≤ 5 y, 18–35 vs. 36–50 years: *p* = 0.465), indicating that certain trauma-related symptoms are less influenced by age. The general lack of significant differences between PTSD ≤ 5 y and PTSD > 5 y groups in most domains and total scores across age strata (e.g., Total Score, 18–50 years: no significant difference) infers that PTSD duration does not substantially alter the overall severity of sleep disturbances, which remain persistently elevated compared to controls, although specific domains show variations by chronicity and age. These highlight sleep disruption as a chronic feature of PTSD with limited resolution over time and underscores the need for targeted interventions across all age groups.

### 3.4. Relationships Between Biomarkers and PSQI-A Domains and Total Score Across Control and PTSD Groups

This analysis examines correlations between four key biomarkers—glutamine (plasma concentrations), glutathione (antioxidant capacity), caspase-1 (inflammatory status), and brain-derived neurotrophic factor (BDNF; neurotrophic support)—and sleep disturbance domains assessed by the PSQI-A across three groups: individuals with a current PTSD diagnosis lasting ≤ 5 years (PTSD ≤ 5 y, N = 33), those with a current PTSD diagnosis lasting > 5 years (PTSD > 5 y, N = 31), and a control group with no diagnosis of PTSD (No PTSD, N = 28).

The PSQI-A evaluates seven domains—hot flashes, general nervousness, trauma-related nightmares, non-trauma-related anxiety, non-trauma-related distressing dreams, episodes of terror, and acting out dreams—along with a total score reflecting overall sleep disruption. By comparing these groups, this section aims to elucidate how biomarker dysregulation correlates with specific sleep disturbances, whether these relationships vary with PTSD chronicity, and how they differ from a non-PTSD population.

Spearman’s rho (ρ) was used to assess non-parametric correlations, with significance determined via two-tailed *p*-values. Correlation matrices were visualized using heatmaps (Figure 2, Figure 3 and Figure 4), with significant correlations (*p* < 0.05) marked by asterisks.

#### 3.4.1. Past PTSD (5 Years or Below)

In the PTSD ≤ 5 y group (current diagnosis lasting 5 years or less), biomarker inter-correlations were moderate, with glutamine–glutathione (ρ = 0.44, *p* = 0.011) weaker than in the No PTSD group, and caspase-1–BDNF showing a weak negative association (ρ = −0.19, *p* = 0.28), while glutamine–BDNF exhibited a moderate negative correlation approaching significance (ρ = −0.31, *p* = 0.083), potentially reflecting interactions between glutamatergic and neurotrophic pathways in recent PTSD (Table 2).

Biomarker–PSQI-A correlations were mostly weak, with glutamine showing significant associations with PSQI-A: Nervousness (ρ = 0.45, *p* = 0.009) and PSQI-A: Non-Trauma Dr (ρ = 0.39, *p* = 0.026), demonstrating that lower glutamine levels may exacerbate these sleep disturbances. BDNF correlations with PSQI-A domains were generally non-significant, such as with PSQI-A: Hot Flashes (ρ = 0.26, *p* = 0.14), indicating limited direct links to sleep disturbances in this group. Other correlations, such as caspase-1 with PSQI-A: Trauma NM (ρ = −0.20, *p* = 0.27), were non-significant.

PSQI-A domains exhibited strong inter-correlations, particularly PSQI-A: Terror Ep with PSQI-A: Dream Act (ρ = 0.66, *p* < 0.001) and both with PSQI-A: Total (ρ = 0.69 and 0.71, *p* < 0.001), indicating their significant contribution to overall sleep disruption. PSQI-A: Trauma NM correlated with Terror Ep and Dream Act (ρ = 0.41, *p* = 0.019 for both), and PSQI-A: Non-Trauma Anx with PSQI-A: Total (ρ = 0.54, *p* = 0.001), reflecting the interconnectedness of trauma- and non-trauma-related sleep symptoms (Figure 2).

Notes: Variables: Biomarkers include glutamine (plasma concentrations, ng/mL), glutathione (antioxidant capacity, nmol/mL), caspase-1 (inflammatory activity, ng/mL), and brain-derived neurotrophic factor (BDNF; neurotrophic support, ng/mL). PSQI-A domains are derived from the Pittsburgh Sleep Quality Index Addendum (PSQI-A). The domains are defined as follows:Hot Flashes: Frequency of hot flashes or sweating episodes during sleep.Nervousness: General feelings of nervousness or restlessness affecting sleep.Trauma NM (Nightmares): Memories or nightmares related to a traumatic event.Non-Trauma Anx (Anxiety): Anxiety or panic episodes during sleep not related to trauma.Non-Trauma Dr (Dreams): Distressing dreams not associated with traumatic events.Terror Ep (Episodes): Episodes of terror or screaming during sleep.Dream Act (Acting out dreams): Physically acting out dreams, such as kicking or thrashing.Total: Sum of all PSQI-A domain scores.

#### 3.4.2. PTSD > 5 y Group

In the PTSD > 5 y group (current diagnosis lasting more than 5 years), biomarkers displayed very strong inter-correlations among glutamine, glutathione, and caspase-1, with glutamine–glutathione (ρ = 0.93, *p* < 0.001), glutamine–caspase-1 (ρ = 0.79, *p* < 0.001), and glutathione–caspase-1 (ρ = 0.80, *p* < 0.001), indicating a highly synchronized biochemical network involving glutamatergic, antioxidant, and inflammatory pathways, possibly due to adaptive responses over time. However, BDNF exhibited weak negative correlations with these biomarkers (e.g., glutamine–BDNF: ρ = −0.12, *p* = 0.52; caspase-1–BDNF: ρ = −0.24, *p* = 0.20), revealing a potential decoupling of neurotrophic support from the other processes in chronic PTSD.

Biomarker–PSQI-A correlations were uniformly weak and non-significant (e.g., glutamine with PSQI-A: Total: ρ = 0.13, *p* = 0.48; caspase-1 with PSQI-A: Trauma NM: ρ = 0.13, *p* = 0.49; BDNF with PSQI-A: Total: ρ = −0.23, *p* = 0.20), indicating minimal direct influence of these biomarkers on sleep disturbances in chronic PTSD. Within PSQI-A domains, strong correlations included PSQI-A: Terror Ep with PSQI-A: Dream Act (ρ = 0.82, *p* < 0.001), PSQI-A: Non-Trauma Anx with PSQI-A: Total (ρ = 0.78, *p* < 0.001), and PSQI-A: Hot Flashes with PSQI-A: Total (ρ = 0.60, *p* < 0.001), underscoring the role of non-trauma-related anxiety and physical sleep behaviors in overall sleep disruption (Figure 3).

Notes: Variables: Biomarkers include glutamine (plasma concentrations, ng/mL), glutathione (antioxidant capacity, nmol/mL), caspase-1 (inflammatory activity, ng/mL), and brain-derived neurotrophic factor (BDNF; neurotrophic support, ng/mL). PSQI-A domains are derived from the Pittsburgh Sleep Quality Index Addendum (PSQI-A). The domains are defined as follows:Hot Flashes: Frequency of hot flashes or sweating episodes during sleep.Nervousness: General feelings of nervousness or restlessness affecting sleep.Trauma NM (Nightmares): Memories or nightmares related to a traumatic event.Non-Trauma Anx (Anxiety): Anxiety or panic episodes during sleep not related to trauma.Non-Trauma Dr (Dreams): Distressing dreams not associated with traumatic events.Terror Ep (Episodes): Episodes of terror or screaming during sleep.Dream Act (Acting out dreams): Physically acting out dreams, such as kicking or thrashing.Total: Sum of all PSQI-A domain scores.

#### 3.4.3. No PTSD Group

In the No PTSD group, biomarkers showed moderate to strong inter-correlations, notably between glutamine and glutathione (ρ = 0.71, *p* < 0.001), indicating a coordinated antioxidant response. Glutamine and BDNF exhibited a moderate negative correlation (ρ = −0.41, *p* = 0.028), suggesting an inverse relationship between glutamatergic and neurotrophic pathways, while caspase-1 and BDNF showed a weak correlation (ρ = −0.06, *p* = 0.78), which was not statistically significant.

Biomarker–PSQI-A correlations were generally weak, with most associations non-significant (e.g., glutamine with PSQI-A: Total: ρ = 0.21, *p* = 0.27). Exceptions included caspase-1 with PSQI-A: Trauma NM (ρ = 0.39, *p* = 0.038), reflecting a potential link between inflammation and trauma-related nightmares even in the absence of PTSD, and BDNF with PSQI-A: Trauma NM (ρ = 0.41, *p* = 0.029), indicating that neurotrophic activity may influence trauma-related sleep disturbances.

Within PSQI-A domains, strong correlations were observed, such as PSQI-A: Hot Flashes with PSQI-A: Terror Ep and PSQI-A: Dream Act (ρ = 1.00, *p* < 0.001), though this likely reflects identical responses due to minimal symptoms in this group. PSQI-A: Nervousness correlated strongly with PSQI-A: Non-Trauma Anx (ρ = 0.60, *p* < 0.001) and PSQI-A: Total (ρ = 0.88, *p* < 0.001), highlighting their contribution to overall sleep quality in controls (Figure 4).

#### 3.4.4. Patterns of Correlations Across Groups

Biomarker inter-correlations showed distinct patterns across groups. For example, the No PTSD group exhibited a Glutamine–Glutathione correlation of ρ = 0.71 (*p* < 0.001), while the corresponding value was ρ = 0.44 (*p* = 0.011) in the PTSD ≤ 5 y group and ρ = 0.93 (*p* < 0.001) in the PTSD > 5 y group. In the PTSD > 5 y group, additional correlations were observed among Glutamine–Caspase-1 (ρ = 0.79, *p* < 0.001) and Glutathione–Caspase-1 (ρ = 0.80, *p* < 0.001). Correlations involving BDNF were generally weak and negative across groups, with Glutamine–BDNF at ρ = −0.41 (*p* = 0.028) in No PTSD, ρ = −0.31 (*p* = 0.083) in PTSD ≤ 5 y, and ρ = −0.12 (*p* = 0.52) in PTSD > 5 y; Caspase-1-BDNF remained non-significant in all groups (ρ = −0.06 to −0.24, *p* > 0.20). Biomarker–PSQI-A correlations also varied: the No PTSD group displayed specific significant associations (e.g., Caspase-1 with Trauma NM: ρ = 0.39, *p* = 0.038; BDNF with Trauma NM: ρ = 0.41, *p* = 0.029), the PTSD ≤ 5 y group showed fewer such links (e.g., Glutamine with Nervousness: ρ = 0.45, *p* = 0.009), and the PTSD > 5 y group had none. PSQI-A inter-correlations were consistently strong across groups, with trauma-related domains (e.g., Trauma NM with Terror Ep: ρ = 0.41, *p* = 0.019) observed in PTSD ≤ 5 y, and non-trauma-related anxiety (ρ = 0.78, *p* < 0.001 with PSQI-A: Total) noted in PTSD > 5 y. These exploratory observations highlight potential variations in association profiles that warrant formal testing in future studies to determine if group differences exist.

Notes: Variables: Biomarkers include glutamine (plasma concentrations, ng/mL), glutathione (antioxidant capacity, nmol/mL), caspase-1 (inflammatory activity, ng/mL), and brain-derived neurotrophic factor (BDNF; neurotrophic support, ng/mL). PSQI-A domains are derived from the Pittsburgh Sleep Quality Index Addendum (PSQI-A). The domains are defined as follows:Hot Flashes: Frequency of hot flashes or sweating episodes during sleep.Nervousness: General feelings of nervousness or restlessness affecting sleep.Trauma NM (Nightmares): Memories or nightmares related to a traumatic event.Non-Trauma Anx (Anxiety): Anxiety or panic episodes during sleep not related to trauma.Non-Trauma Dr (Dreams): Distressing dreams not associated with traumatic events.Terror Ep (Episodes): Episodes of terror or screaming during sleep.Dream Act (Acting out dreams): Physically acting out dreams, such as kicking or thrashing.Total: Sum of all PSQI-A domain scores.

## 4. Discussion

Post-Traumatic Stress Disorder is a complex and heterogeneous condition that arises as a result of severe psychological trauma [40]. It can persist for extended periods and is associated with numerous biological consequences. On one hand, stress can initiate a cascade of biochemical processes aimed at restoring physiological balance and supporting recovery. However, a far more serious outcome is allostatic overload, a condition reflecting chronic stress-related imbalance that disrupts the body’s homeostasis, weakens internal regulatory and defense mechanisms, and can ultimately lead to disease development [41,42]. PTSD is increasingly viewed as a form of pathological adaptation to chronic traumatic stress, resulting in long-term changes within the nervous, endocrine, and immune systems [42]. Understanding the biological foundations of PTSD is a significant focus of modern neuropsychiatric research [43]. The reclassification of PTSD in the Diagnostic and Statistical Manual of Mental Disorders, Fifth Edition (DSM-5)—from an anxiety disorder to the newly created category of Trauma- and Stressor-Related Disorders—reflects growing clinical recognition that its core features include not only anxiety symptoms but also distinct neurobiological alterations. These changes involve brain structures responsible for memory, emotional processing, and stress regulation.

Chronic PTSD is linked to progressive deterioration of brain regions involved in emotion and stress regulation, particularly the amygdala, hippocampus, prefrontal cortex, and HPA axis [44,45,46]. In this context, abnormal amygdala hyperactivity disrupts fear learning and impairs emotional regulation, contributing to heightened fear responses, chronic hypervigilance, and symptoms such as intrusive thoughts and persistent anxiety [47].

Our research involved individuals working under conditions of extreme stress and high trauma exposure risk. The study groups included participants with a current PTSD diagnosis lasting ≤ 5 years (PTSD ≤ 5 y), those with a current PTSD diagnosis lasting > 5 years (PTSD > 5 y), and a control group with no diagnosis of PTSD (No PTSD). The groups did not differ significantly in age or duration of employment, suggesting that observed biomarker differences were attributable to PTSD status. Additionally, individuals with PTSD exhibited co-occurring behavioral addictions, including disordered eating, compulsive visualization, and exercise addiction. Notably, the PTSD > 5 y group showed the longest-lasting behavioral compulsions.

These findings suggest that certain behavioral patterns—such as compulsive eating, visual intrusions, and excessive exercise—can be classified as non-substance-related addictive behaviors consistent with DSM-5 criteria and within the spectrum of impulse-control disorders. While not chemically mediated, these behaviors may nonetheless influence biological systems involved in stress regulation, oxidative balance, and sleep architecture. Given their high prevalence in PTSD populations and their potential overlap with neuroinflammatory and excitatory processes, these behaviors warrant further investigation—particularly in studies exploring addiction–biomarker interactions.

It should be acknowledged that the relatively modest sample size of 92 subjects divided into three groups and further stratified by age may limit the statistical power of some subgroup comparisons. This limitation warrants caution in the interpretation of findings and should be considered when drawing conclusions. Larger cohorts in future studies would help to confirm and extend these results. To enhance the robustness and generalizability of results, future studies should aim to include larger sample sizes, which would facilitate the application of more sophisticated statistical models and appropriate control for confounding variables. From a neurobiological perspective, expanding the study population would allow for a more precise characterization of the dynamic changes in biomarkers associated with the pathophysiology of PTSD, thereby deepening the understanding of the molecular and cellular mechanisms underlying the disorder and aiding the identification of potential therapeutic targets.

This study analyzed four biomarkers: glutamine, glutathione, caspase-1 concentration and BDNF levels—reflecting neurochemical, inflammatory, oxidative, and neurotrophic changes associated with PTSD.

Plasma concentrations of glutamine, glutathione, caspase-1 and BDNF differed significantly among the three groups (*p* < 0.001). Glutamine and GSH were lowest in PTSD ≤5 y and highest in the control group, indicating impaired excitatory–inhibitory balance and antioxidant defenses in PTSD

Glutamine, as a precursor of glutamate, is central to maintaining the excitation–inhibition balance. Its dysregulation may promote excitotoxicity and impaired neuroplasticity, which in PTSD can manifest as difficulty extinguishing fear or suppressing traumatic memories [48]. Glutathione, the body’s primary antioxidant, protects the brain from oxidative stress. Reduced GSH levels in PTSD patients suggest increased free radical production and compromised antioxidant defenses. GSH depletion may therefore serve as a biomarker of oxidative stress in PTSD and indicate neuronal injury [49].

Other studies using magnetic resonance spectroscopy (MRS) have confirmed alterations in glutamine and glutamate concentrations in regions such as the hippocampus, prefrontal cortex, and amygdala—areas vital for emotion regulation, memory, and stress response [50,51]. For instance, Sheth C. et al. found reduced GABA and glutamine in the anterior cingulate cortex (ACC) in trauma-exposed veterans [52]. Similarly, Yang Z.Y. et al. observed reduced glutamine in the ACC of children diagnosed with PTSD following the 2008 Wenchuan earthquake [53]. Watling S.E. et al. reported decreased GSH levels and increased malondialdehyde (MDA)—a lipid peroxidation product—in terrorism survivors [10]. Wiercioch W. et al. found decreased serum GSH in PTSD patients, with a correlation between low GSH and greater PTSD symptom severity [54]. Maier A. et al. examined the effect of N-acetylcysteine (NAC), a GSH precursor, noting that supplementation may raise brain GSH levels and potentially alleviate PTSD symptoms [55]. Consistently, Hasan H.M. demonstrated that women with PTSD living in the Kurdistan Region of Iraq exhibited significantly decreased levels of reduced glutathione (GSH), reinforcing the notion that oxidative stress and impaired antioxidant defenses are central features of PTSD pathophysiology [56].

Caspase-1, also known as interleukin-1β converting enzyme, is involved in the inflammatory response and activates pro-inflammatory cytokines such as IL-1β. Elevated caspase-1 concentration level in PTSD may indicate inflammasome activation and chronic neuroinflammation [24]. It may act as a neuroinflammatory marker and influence mood and behavior through cytokine pathways. In the study by Perez-Barcena J.M., higher caspase-1 levels in serum at hospital admission predicted worse outcomes six months post-traumatic brain injury (TBI) [57]. Liu W. et al. showed that mechanical stress from traumatic brain injury (TBI) activates caspase-1 and contributes to pyroptosis, a form of inflammatory cell death [58]. PTSD is characterized by persistent fear responses and dysregulated neurotransmission. Stress particularly affects glutamate, a neurotransmitter essential for synaptic plasticity and memory formation [59,60,61].

BDNF levels were reduced in both PTSD groups, with the lowest values in chronic PTSD (>5 years). This pattern may reflect progressive neuroplastic dysfunction, consistent with previous findings of reduced BDNF in the hippocampus and prefrontal cortex [16,17,18]. Such deficits are linked to impaired extinction of traumatic associations and persistence of intrusive symptoms [19].

In our sample, we did not observe strong correlations between BDNF and PSQI-A domains within PTSD groups, which may suggest that the contribution of BDNF to sleep disturbances may be indirect and modulated by other processes, including oxidative stress and neuroinflammation. Interestingly, in the control group, BDNF was significantly associated with trauma-related nightmare frequency, highlighting that even in the absence of clinical PTSD, neurotrophic mechanisms might influence vulnerability to trauma-linked sleep disturbances. From a therapeutic perspective, our findings are in line with the literature proposing a potential role of BDNF as both a biomarker and a potential target in PTSD. Interventions known to enhance BDNF signaling—such as physical exercise, certain antidepressants, or neuromodulatory approaches—could potentially help restore neural plasticity and improve sleep and emotional regulation [20]. The sustained suppression of BDNF in chronic PTSD may point toward the need for treatment strategies that address not only acute symptoms but also long-term neuroplastic deficits that may perpetuate the disorder. Taken together, the analysis of all four biomarkers—glutamine, glutathione, caspase-1, and BDNF—indicates that PTSD is underpinned by a combination of excitotoxic, oxidative, inflammatory, and neurotrophic alterations. These processes interact and may contribute to the persistence of symptoms, including impaired memory processing, emotional dysregulation, and sleep disturbances.

In our study, participants with PTSD scored significantly worse on the PSQI-A compared to those without PTSD. Symptoms such as hot flashes worsened with age and were more prevalent in the PTSD ≤ 5 y group (current diagnosis lasting 5 years or less) than in the PTSD > 5 y group (current diagnosis lasting more than 5 years). While biomarker associations with overall sleep disturbance (as measured by the PSQI-A) were generally weak in the PTSD ≤ 5 y group, glutamine levels showed significant correlations with specific PSQI-A subdomains. In the PTSD > 5 y group, strong inter-correlations emerged across several PSQI-A domains, underscoring the ongoing influence of non-trauma-related anxiety and physiological arousal on persistent sleep disturbances in chronic PTSD.

Despite the valuable insights from this study, several limitations must be acknowledged. The modest sample size (n = 92), while sufficient for detecting moderate effect sizes, restricts statistical power—particularly in subgroup and age-stratified analyses—reducing sensitivity to group-by-age interactions, subtle biomarker variations, and complex multivariate modeling. This may introduce residual bias from unadjusted confounders, such as trauma type, medication status, or comorbidities, and limits the generalizability of findings. Additionally, the exploratory nature of the correlation analyses renders interpretations preliminary and hypothesis-generating, necessitating validation in larger confirmatory studies with multiplicity adjustments. The exclusive focus on male participants further constrains insights into sex-specific differences in PTSD pathophysiology, biomarker profiles, and sleep disturbances, which may be influenced by hormonal, neurobiological, or psychosocial factors in women.

Future research should investigate targeted interventions for biomarker dysregulations, including antioxidant therapies to boost glutathione levels or BDNF-enhancing approaches (e.g., exercise, selective serotonin reuptake inhibitors, or neuromodulation) to improve sleep and symptom management in PTSD. Longitudinal designs are recommended to trace biomarker and sleep trajectories, clarifying causal links and chronicity-related progression. Incorporating female participants in sex-comparative studies would address the male-only limitation and explore potential sex-based variations in pathophysiology and treatment responses.

## 5. Conclusions

The observed increase in inter-biomarker correlations with advancing PTSD chronicity suggests a dynamic neurobiological adaptation in which inflammatory, glutamatergic, and oxidative systems become progressively interconnected, potentially reflecting a compensatory mechanism to preserve neural homeostasis. These findings emphasize the evolving complexity of PTSD pathogenesis, characterized by integration of neuroimmune and excitatory neurotransmission pathways. Despite these molecular changes, associations with PSQI-A scores remained weak across groups, indicating that sleep disturbances in PTSD are more strongly influenced by psychological and cognitive–emotional factors, as shown by robust intercorrelations within PSQI-A domains. We also observed a transition from trauma-specific sleep disturbances in early PTSD (≤5 years) to more generalized impairments in chronic PTSD (>5 years).

This highlights the need for adaptive treatment strategies that evolve with clinical phenotype, including supplementation of trauma-focused therapies with interventions targeting generalized anxiety and circadian rhythm disruption. Notably, discrete associations between select biomarkers and PSQI-A subdomains were detected even in controls (e.g., caspase-1 with trauma-related nightmares), suggesting that subclinical neurobiological vulnerabilities may be predisposed to sleep dysregulation. Overall, these results support the rationale for integrated, personalized treatment strategies that simultaneously address excitatory neurotransmission and inflammasome-mediated inflammation. Although the therapeutic promise of biomarker-informed approaches is considerable, clinical implementation remains limited, underscoring the need for further translational and clinical research to validate their efficacy and safety.

The progressive strengthening of inter-correlations among glutamine, glutathione, and caspase-1 with PTSD chronicity emphasizes an evolving biological adaptation, where glutamatergic, antioxidant, and inflammatory pathways become more synchronized over time, potentially as a compensatory mechanism. In contrast, the weakening inverse associations involving BDNF suggest a diminishing interaction between neurotrophic support and these pathways in chronic cases. However, the consistent lack of strong biomarker–PSQI-A correlations across all groups indicates that these biomarkers may not directly drive sleep disturbances in PTSD, which appear more closely tied to psychological factors, as evidenced by the robust PSQI-A inter-correlations. The shift from trauma-related to non-trauma-related sleep disturbances in chronic PTSD (>5 y) highlights the need for tailored interventions that address evolving symptom profiles, such as targeting non-trauma anxiety in long-standing cases. The specific biomarker–PSQI-A associations in the No PTSD group (e.g., caspase-1 and BDNF with Trauma NM) indicate subclinical pathways that may predispose individuals to sleep issues, warranting further investigation.

## Figures and Tables

**Figure 1 cimb-47-00814-f001:**
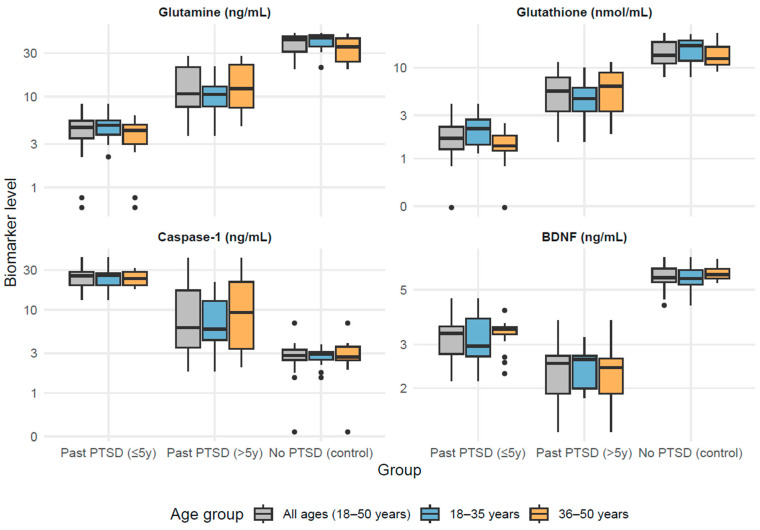
Biomarker levels across PTSD groups by age distribution.

**Figure 2 cimb-47-00814-f002:**
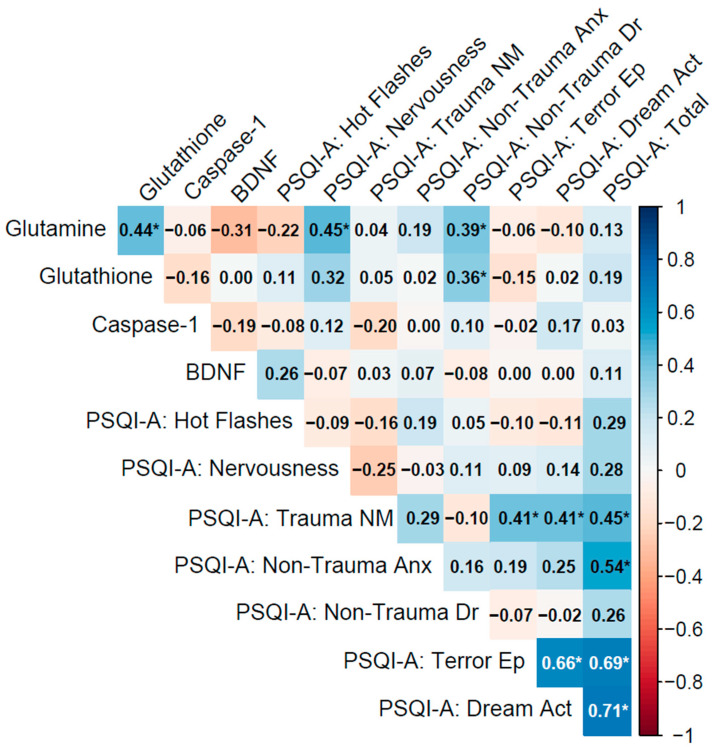
Spearman correlation matrix of biomarkers (glutamine, glutathione, caspase-1, BDNF) and PSQI-A scores in individuals with a current PTSD diagnosis lasting less than 5 years (n = 33).

**Figure 3 cimb-47-00814-f003:**
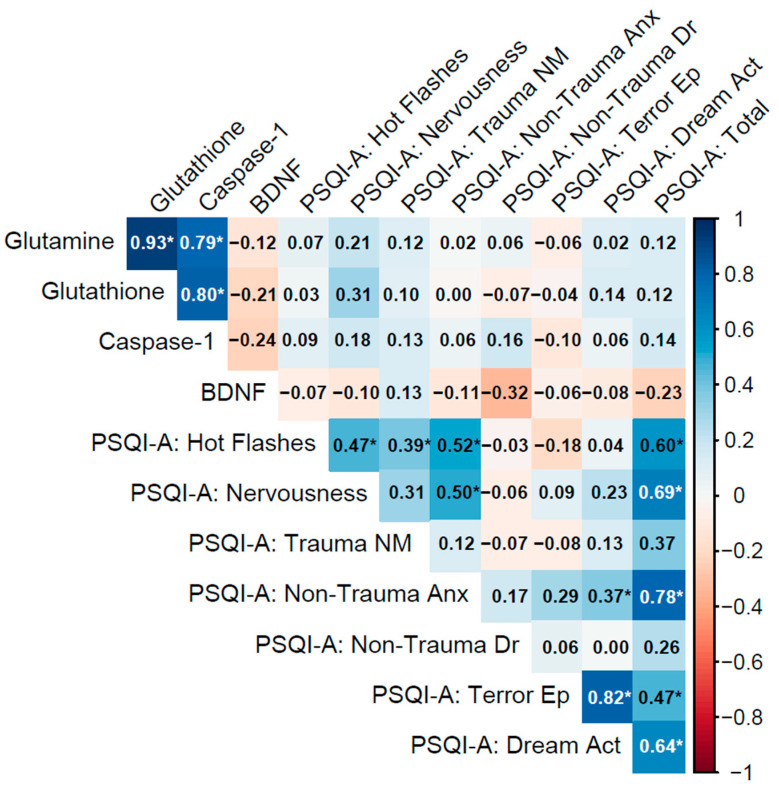
Spearman correlation matrix of biomarkers (glutamine, glutathione, caspase-1, BDNF) and PSQI-A scores in individuals with a current PTSD diagnosis lasting more than 5 years (n = 31).

**Figure 4 cimb-47-00814-f004:**
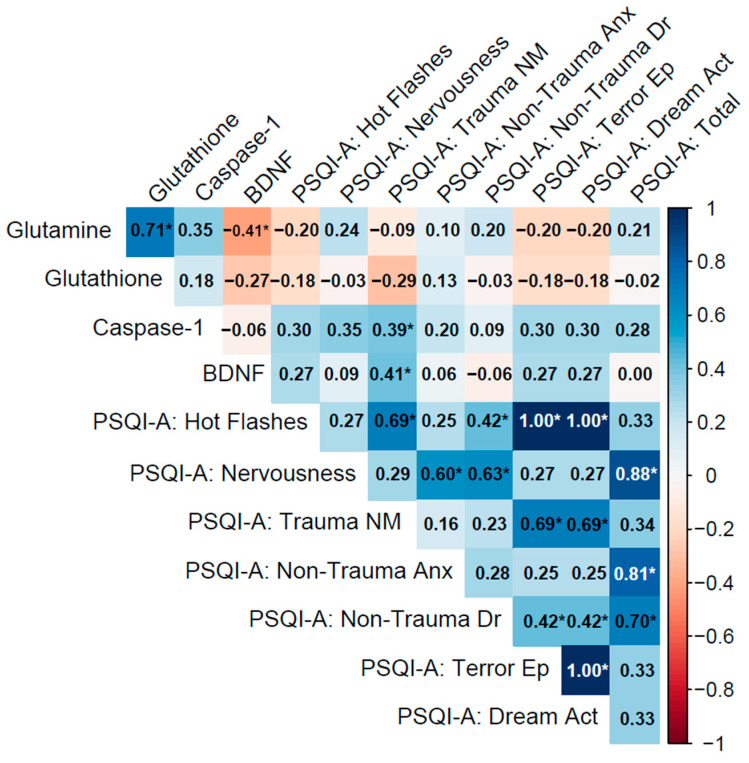
Spearman correlation matrix of biomarkers (glutamine, glutathione, caspase-1, BDNF) and PSQI-A scores in the No PTSD group (n = 28).

**Table 1 cimb-47-00814-t001:** Demographic and addiction-related characteristics of participants by PTSD status.

Characteristic	Total (N = 92)	PTSD ≤ 5 y (N = 33)	PTSD > 5 y (N = 31)	No PTSD (Control) (N = 28)	*p*
Demographic characteristics					
Age (years)	34.0 (28.8–41.0)	34.0 (31.0–41.0)	36.0 (29.5–41.0)	33.5 (24.3–41.5)	0.524
Employment in hazardous conditions (years)	10.0 (6.0–14.3)	11.0 (7.0–14.0)	10.0 (7.5–15.0)	10.0 (3.0–14.0)	0.418
BMI (kg/m^2^)	23.0 (20.8–25.0)	22.0 ^a^ (22.0–24.0)	22.0 ^a^ (20.0–24.5)	25.0 ^b^ (22.0–28.3)	0.011
Addiction-related characteristics					
Compulsive behavioral addictions (years)	1.0 (0.0–5.0)	1.0 ^b^ (0.0–5.0)	4.0 ^c^ (3.0–5.0)	0.0 ^a^ (0.0–1.0)	<0.001
Eating-related addictions (years)	2.0 (0.0–4.0) (N = 91)	3.0 ^b^ (2.0–5.0)	3.0 ^b^ (1.5–4.5)	0.0 ^a^ (0.0–1.0)	<0.001
Pathological visualization disorders (years)	1.0 (0.0–3.0) (N = 91)	1.0 ^b^ (0.0–5.0)	2.0 ^b^ (0.5–3.5)	0.0 ^a^ (0.0–1.0)	0.004
Cyberaddictions (years)	0.0 (0.0–1.0) (N = 91)	0.0 (0.0–1.0)	1.0 (0.0–2.5)	0.0 (0.0–1.5)	0.197
Exercise compulsion addiction (years)	1.0 (0.0–2.0) (N = 90)	1.0 ^b^ (0.0–3.0)	1.5 ^b^ (0.0–2.0)	0.0 ^a^ (0.0–1.0)	<0.001
CAPS-5 total score	57.5 (10.5–67.0)	68.0 ^c^ (66.0, 72.0)	56.00 ^b^ (54.0, 59.0)	7.0 ^a^ (4.5, 9.0)	<0.001

Notes: Data are presented as median (interquartile range, IQR) unless otherwise specified. All *p*-values were determined using the Kruskal–Wallis rank sum test. Superscripts (a, b, c) indicate statistically distinct groups based on post hoc pairwise comparisons (e.g., Dunn’s test with Bonferroni correction); groups sharing the same letter do not differ significantly (*p* > 0.05). BMI—body mass index; PTSD = post-traumatic stress disorder.

**Table 2 cimb-47-00814-t002:** Biomarker levels across PTSD groups by age distribution.

Characteristic	Age Group	N	PTSD ≤ 5 y (N = 33)	PTSD > 5 y (N = 31)	No PTSD (Control) (N = 28)	*p*
Glutamine levels (ng/mL)	18–50 yrs	92	4.55 ^a^ (3.46, 5.40)	10.72 ^b^ (7.70, 21.07)	42.52 ^c^ (31.18, 45.68)	<0.001
	18–35 yrs	45	4.83 ^a^ (3.80, 5.47)	10.54 ^b^ (7.75, 12.98)	44.30 ^c^ (35.67, 47.03)	<0.001
	36–50 yrs	47	4.22 ^a^ (3.00, 4.92)	12.15 ^b^ (7.60, 22.31)	35.26 ^c^ (24.06, 43.58)	<0.001
Glutathione levels (nmol/mL)	18–50 yrs	92	1.67 ^a^ (1.27, 2.24)	5.52 ^b^ (3.33, 7.89)	13.69 ^c^ (11.19, 19.21)	<0.001
	18–35 yrs	45	2.15 ^a^ (1.44, 2.71)	4.58 ^b^ (3.33, 6.03)	17.55 ^c^ (11.93, 19.92)	<0.001
	36–50 yrs	47	1.38 ^a^ (1.22, 1.80)	6.32 ^b^ (3.38, 8.86)	12.61 ^c^ (10.83, 17.05)	<0.001
Caspase-1 levels (ng/mL)	18–50 yrs	92	25.70 ^c^ (19.72, 28.57)	6.09 ^b^ (3.53, 17.23)	2.77 ^a^ (2.40, 3.29)	<0.001
	18–35 yrs	45	25.92 ^c^ (19.74, 27.80)	5.87 ^b^ (4.38, 12.75)	2.94 ^a^ (2.51, 3.11)	<0.001
	36–50 yrs	47	23.64 ^b^ (19.60, 28.57)	9.37 ^b^ (3.40, 21.62)	2.66 ^a^ (2.14, 3.59)	<0.001
BDNF levels (ng/mL)	18–50 yrs	92	3.33 ^b^ (2.75, 3.56)	2.52 ^a^ (1.88, 2.73)	5.61 ^c^ (5.37, 6.10)	<0.001
	18–35 yrs	45	2.96 ^b^ (2.69, 3.90)	2.61 ^a^ (1.88, 2.73)	5.56 ^c^ (5.23, 6.09)	<0.001
	36–50 yrs	47	3.46 ^b^ (3.31, 3.54)	2.43 ^a^ (1.89, 2.71)	5.76 ^c^ (5.55, 6.11)	<0.001

Notes: Median (IQR): Data are presented as median values with interquartile ranges (Q1, Q3). Glutamine levels represent plasma concentrations, glutathione (GSH) levels indicate antioxidant capacity, caspase-1 levels reflect inflammatory status, and brain-derived neurotrophic factor (BDNF) measures neurotrophic support. *p*-value: Calculated using Quade’s rank-based analysis of covariance (ANCOVA) to account for potential confounding by BMI to assess differences across the three groups (PTSD ≤ 5 y, PTSD > 5 y, No PTSD) within each age category. A *p* < 0.001 indicates statistically significant differences at the α = 0.05 level, adjusted for multiple comparisons. Superscripts (a, b, c) indicate statistically distinct groups based on post hoc pairwise comparisons (e.g., Quade’s test with Bonferroni correction); groups sharing the same letter do not differ significantly (*p* > 0.05). Sample Size (N): Refers to the number of participants with available data for each biochemical parameter within the specified age group. The total cohort (N = 92) was divided into PTSD ≤ 5 years (N = 33), PTSD > 5 years (N = 31), and No PTSD (N = 28) for the 18–50 years group, with corresponding subgroups for 18–35 and 36–50 years. Clinical context: PTSD ≤ 5 y denotes individuals with a current PTSD diagnosis lasting 5 years or less, PTSD > 5 y indicates a current PTSD diagnosis lasting more than 5 years, and No PTSD serves as the control group with no diagnosis of PTSD.

**Table 3 cimb-47-00814-t003:** PSQI-A scores across control and PTSD groups by age subgroup.

Domain	Age Subgroup	N	PTSD ≤ 5 y (N = 33)	PTSD > 5 y (N = 31)	No PTSD (Control) (N = 28)	*p*
Hot flashes	18–50 yrs	92	2.55 (0.67) ^c^	2.13 (0.92) ^b^	0.07 (0.38) ^a^	<0.001
	18–35 yrs	45	2.75 (0.45) ^c^	1.60 (0.91) ^b^	0.00 (0.00) ^a^	<0.001
	36–50 yrs	47	2.35 (0.79) ^b^	2.63 (0.62) ^b^	0.14 (0.53) ^a^	<0.001
General nervousness	18–50 yrs	92	2.45 (0.62) ^b^	2.42 (0.67) ^b^	0.68 (0.94) ^a^	<0.001
	18–35 yrs	45	2.38 (0.62) ^b^	2.20 (0.68) ^b^	0.79 (0.89) ^a^	<0.001
	36–50 yrs	47	2.53 (0.62) ^b^	2.63 (0.62) ^b^	0.57 (1.02) ^a^	<0.001
Memories/nightmares of trauma	18–50 yrs	92	2.82 (0.46) ^b^	2.87 (0.34) ^b^	0.14 (0.52) ^a^	<0.001
	18–35 yrs	45	2.94 (0.25) ^b^	2.80 (0.41) ^b^	0.00 (0.00) ^a^	<0.001
	36–50 yrs	47	2.71 (0.59) ^b^	2.94 (0.25) ^b^	0.29 (0.73) ^a^	<0.001
Anxiety/panic not related to trauma	18–50 yrs	92	2.67 (0.54) ^b^	2.48 (0.77) ^b^	1.11 (0.79) ^a^	<0.001
	18–35 yrs	45	2.81 (0.40) ^b^	2.40 (0.83) ^b^	1.29 (0.73) ^a^	<0.001
	36–50 yrs	47	2.53 (0.62) ^b^	2.56 (0.73) ^b^	0.93 (0.83) ^a^	<0.001
Distressing dreams not related to trauma	18–50 yrs	92	2.70 (0.47) ^b^	2.65 (0.55) ^b^	0.43 (0.84) ^a^	<0.001
	18–35 yrs	45	2.19 (0.40) ^b^	2.33 (0.49) ^b^	0.00 (0.00) ^a^	<0.001
	36–50 yrs	47	2.65 (0.49) ^b^	2.75 (0.58) ^b^	0.29 (0.83) ^a^	<0.001
Episodes of terror	18–50 yrs	92	2.33 (0.54) ^b^	2.13 (0.56) ^b^	0.11 (0.57) ^a^	<0.001
	18–35 yrs	45	2.19 (0.40) ^b^	2.33 (0.49) ^b^	0.00 (0.00) ^a^	<0.001
	36–50 yrs	47	2.47 (0.62) ^c^	1.94 (0.57) ^b^	0.21 (0.80) ^a^	<0.001
Acting out dreams	18–50 yrs	92	2.48 (0.51) ^c^	2.00 (0.73) ^b^	0.11 (0.57) ^a^	<0.001
	18–35 yrs	45	2.50 (0.52) ^b^	2.07 (0.80) ^b^	0.00 (0.00) ^a^	<0.001
	36–50 yrs	47	2.47 (0.51) ^c^	1.94 (0.68) ^b^	0.21 (0.80) ^a^	<0.001
Total score	18–50 yrs	92	18.00 (1.80) ^c^	16.68 (2.55) ^b^	2.64 (3.39) ^a^	<0.001
	18–35 yrs	45	18.31 (1.62) ^c^	15.93 (2.28) ^b^	2.64 (1.91) ^a^	<0.001
	36–50 yrs	47	17.71 (1.96) ^b^	17.38 (2.66) ^b^	2.64 (4.50) ^a^	<0.001

Notes: Mean (SD): Data are presented as mean values with standard deviation. *p*-value: Calculated using the ANCOVA test with BMI as covariate to assess differences across the three groups (PTSD ≤ 5 y, PTSD > 5 y, No PTSD) within each age category. A *p* < 0.001 indicates statistically significant differences at the α = 0.05 level. ABC Notation: Letters (a, b, c) indicate significant differences based on the overall test using compact letter display design. Letters are assigned in ascending alphabetical order corresponding to increasing group mean values, where ‘a’ denotes the minimum (lowest) value, and higher letters like ‘c’ (if applicable) denote the maximum (highest) value. Groups sharing the same letter do not differ significantly. For post hoc test the Games–Howell test was conducted with Holm–Bonferroni correction for multiple comparisons. Sample Size (N): Refers to the number of participants with available data for each biochemical parameter within the specified age group. The total cohort (N = 92) was divided into PTSD ≤ 5 years (N = 33), PTSD >5 years (N = 31), and No PTSD (N = 28) for the 18–50 years group, with corresponding subgroups for 18–35 and 36–50 years. Clinical context: PTSD ≤ 5 y denotes individuals with a current PTSD diagnosis lasting 5 years or less, PTSD > 5 y indicates a current PTSD diagnosis lasting more than 5 years, and No PTSD serves as the control group with no diagnosis of PTSD.

## Data Availability

All data and analysis are available within the manuscript, or upon request to the corresponding author.

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
