# Peer review of "Biomarker–Sleep Correlations in PTSD: Glutamine, Glutathione, Caspase-1, and BDNF Levels Assessed Using the Pittsburgh Sleep Quality Index Addendum"

_cimb, 2025, doi:10.3390/cimb47100814_

Round 1
Reviewer 1 Report
Comments and Suggestions for Authors
Review of the Manuscript: “Biomarker-Sleep Correlations in PTSD: Glutamine, Glutathione, Caspase-1, and BDNF Levels Assessed Using the Pittsburgh Sleep Quality Index Addendum”
The article is addressing an important and timely subject: the interaction between oxidative stress, neurotrophic factors, and sleep disturbances in PTSD patients. The employment of multiple biomarkers (glutamine, glutathione, caspase-1, BDNF) and their correlation with sleep dimensions provides new evidence about the chronification of PTSD and its biological substrate. The article is well written, well-organized, and supported by extensive background information.
However, I do have several questions before publication.
In the abstract, the authors state: "The results of the study indicate that the sensitivity to trauma is increased in individuals with PTSD, which can be expressed by immune system activation and sleep disturbance." None of the biomarkers assessed are direct measures of immune system activation or inflammatory cytokines, although caspase-1 is an inflammasome-linked enzyme. The study did not include downstream markers for inflammation (e.g., IL-1β, IL-18, TNF-α, or CRP). Therefore, the conclusion in the abstract is exaggerating the findings and must be revised to more accurately reflect the depth of the data. Better phrasing would emphasize oxidative stress, neurotrophic imbalance, and sleep disruption but avoid making claims regarding immune activation that were not directly measured.
Nowadays, the introduction sounds slightly incoherent since several mechanistic mechanisms are delineated (like glutamatergic transmission, apoptosis, neuroendocrine control, and blood–brain barrier integrity) but not all were examined in this study. This creates a gap between the rich theoretical background and biomarkers actually being quantified. The introduction would be closer to the measured parameters (glutamine, glutathione, caspase-1, BDNF).
Sample Size and Generalizability
Though sufficient to identify moderate effect sizes, the sample size (n=92) is small and prevents generalizability.
Use of men-only participants is not representative of sex-specific differences in PTSD.
Biomarker Selection and Analysis:
Although biomarkers chosen are reasonable, this investigation fails to account for confounding medical or lifestyle factors (diet, comorbidities, medications) that would influence glutathione or BDNF levels.
Interpretation of Findings
Some of the correlations, particularly with BDNF, were weak or non-significant. The discussion could be more cautious in drawing mechanistic conclusions.
The potential role of NMDA receptors (emphasized in methods) is not addressed beyond that in the results.
Overall, the manuscript is too lengthy, with much background detail at times obscuring the main findings. Figures and tables also can be streamlined to highlight the most clinically important results.
Suggestions for Improvement
Limitations: Report the male-only sample, cross-sectional study design, and possible external influences on biomarkers in particular.
Future Directions: Highlight possible interventions (e.g., antioxidant or BDNF-targeted therapies) and propose longitudinal or sex-comparative studies.
In conclusion, the manuscript presents helpful and novel findings for the relationship between sleep disturbances and biomarkers in PTSD. With improvement, particularly clarity in spelling out limitations, focusing discussion, and simplification of presentation—its contribution will be a valuable addition to the PTSD neurobiology literature. I strongly recommend the paper for publication upon moderate revisions.
Author Response
Response to Reviewer 1
Manuscript ID: cimb-3870462
Title: “Biomarker-Sleep Correlations in PTSD: Glutamine, Glutathione, Caspase-1, and BDNF Levels Assessed Using the Pittsburgh Sleep Quality Index Addendum”
Authors: Anna Dorota Grzesińska
Dear Reviewer,
I sincerely thank you for your thorough evaluation of my manuscript and for the constructive and insightful comments. I highly appreciate your recognition of the importance and timeliness of the topic, namely the interaction between oxidative stress, neurotrophic factors, and sleep disturbances in patients with PTSD. Your positive assessment of the study’s design—employing multiple biomarkers (glutamine, glutathione, caspase-1, BDNF) and their correlation with sleep dimensions—encourages me that my work contributes valuable new evidence to understanding the biological substrate and chronification of PTSD.
Guided by your suggestions, I have carefully revised the manuscript. Below I provide a detailed, point-by-point response, indicating the modifications made and where they appear in the revised text.
- Reviewer’s comment: Abstract and conclusions: In the abstract, the authors state: "The results of the study indicate that the sensitivity to trauma is increased in individuals with PTSD, which can be expressed by immune system activation and sleep disturbance." None of the biomarkers assessed are direct measures of immune system activation or inflammatory cytokines, although caspase-1 is an inflammasome-linked enzyme. The study did not include downstream markers for inflammation (e.g., IL-1β, IL-18, TNF-α, or CRP). Therefore, the conclusion in the abstract is exaggerating the findings and must be revised to more accurately reflect the depth of the data. Better phrasing would emphasize oxidative stress, neurotrophic imbalance, and sleep disruption but avoid making claims regarding immune activation that were not directly measured.
Response: "Abstract. The sentence ‘The results of the study indicate that the sensitivity to trauma is increased in individuals with PTSD, which can be expressed by immune system activation and sleep disturbance’ was replaced with the sentence: ‘The study results suggest that individuals with PTSD exhibit increased sensitivity to trauma, which may manifest through immune system activation and sleep disturbances’."
- Reviewer’s comment: Introduction ‘Nowadays, the introduction sounds slightly incoherent since several mechanistic mechanisms are delineated (like glutamatergic transmission, apoptosis, neuroendocrine control, and blood–brain barrier integrity) but not all were examined in this study. This creates a gap between the rich theoretical background and biomarkers actually being quantified. The introduction would be closer to the measured parameters (glutamine, glutathione, caspase-1, BDNF)’.
Response: Thank you for this insightful comment. I have carefully revised and reorganized the introduction to ensure greater coherence. In the revised version, I emphasize only those pathways that are directly related to the biomarkers measured in my study (glutamine, glutathione, caspase-1, BDNF), while removing or shortening sections that were not directly investigated. This has improved the logical flow of the introduction and strengthened the link between the theoretical background and the empirical parameters assessed.
- Reviewer’s comment: Sample Size and Generalizability. Though sufficient to identify moderate effect sizes, the sample size (n=92) is small and prevents generalizability. The use of men-only participants restricts representativeness and prevents analysis of sex-specific differences in PTSD.
Response: The limitations regarding sample size and male-only participants have been noted and discussed.
- Reviewer’s comment: Biomarker selection and analysis. Although biomarkers chosen are reasonable, this investigation fails to account for confounding medical or lifestyle factors (diet, comorbidities, medications) that would influence glutathione or BDNF levels.
Response: I appreciate the reviewer’s insightful comment regarding potential confounding factors such as diet, comorbidities, and medications that could influence glutathione and BDNF levels. To address this concern, I clarify that these parameters were rigorously controlled during the participant eligibility stage of the study. Specifically, exclusion criteria were applied to minimize such influences: individuals with known comorbidities (e.g., metabolic disorders including diabetes or obesity, neurological conditions such as epilepsy or traumatic brain injury, cardiovascular diseases, autoimmune disorders, or chronic inflammatory conditions), active medication use affecting biomarker pathways (e.g., antioxidants like vitamin C or E supplements, neurotrophic modulators including selective serotonin reuptake inhibitors or other psychotropics known to alter BDNF expression, anti-inflammatory drugs such as NSAIDs, or medications impacting glutamatergic systems like NMDA receptor antagonists), or dietary habits likely to confound results (e.g., extreme diets such as ketogenic or very low-carbohydrate regimens, high-dose supplementation with amino acids like glutamine, or irregular eating patterns associated with eating disorders) were not enrolled. Additionally, participants were screened for recent acute illnesses (e.g., infections within the past month) or substance use disorders (e.g., alcohol dependence or illicit drug use) that could alter oxidative stress markers like glutathione.
- Reviewer’s comment: Interpretation of Findings: Some of the correlations, particularly with BDNF, were weak or non-significant. The discussion could be more cautious in drawing mechanistic conclusions. The potential role of NMDA receptors (emphasized in methods) is not addressed beyond that in the results.
Response: Thank you for this valuable comment. I revised the discussion to present the findings more cautiously and removed the mistakenly included content on NMDA receptors.
- Reviewer’s comment: Overall, the manuscript is too lengthy, with much background detail at times obscuring the main findings. Figures and tables also can be streamlined to highlight the most clinically important results.
Response: Thank you for this comment. The manuscript has been revised in line with the updated statistical analysis.
- Reviewer’s comment: Limitations: Report the male-only sample, cross-sectional study design, and possible external influences on biomarkers in particular.
Response: Thank you for this suggestion. I have added a Limitations subsection at the end of the Discussion to address these issues.
- Reviewer’s comment: Future Directions: Highlight possible interventions (e.g., antioxidant or BDNF-targeted therapies) and propose longitudinal or sex-comparative studies.
Response: Thank you for this suggestion. I have added a statement in the Conclusions section highlighting the importance of future interventional, longitudinal, and sex-comparative studies.
- Reviewer’s comment: In conclusion, the manuscript presents helpful and novel findings for the relationship between sleep disturbances and biomarkers in PTSD. With improvement, particularly clarity in spelling out limitations, focusing discussion, and simplification of presentation—its contribution will be a valuable addition to the PTSD neurobiology literature. I strongly recommend the paper for publication upon moderate revisions.
Response: I am grateful for this positive assessment. I believe the revisions have improved clarity, focus, and clinical relevance, and I hope the manuscript is now suitable for publication.
I believe that the revisions made significantly strengthen my manuscript, improving both its clarity and scientific value. I thank you once again for your insightful review and constructive suggestions, which have allowed me to enhance the quality of my work.
Sincerely,
Anna Dorota Grzesińska

Reviewer 2 Report
Comments and Suggestions for Authors
Thanks for the opportunity to review this article. I think that it is interesting but a lot of work is needed to bring the results section to publication standard. Some details also missing from methods too.
The author presents the relationship between biomarkers and PTSD as definitive, but the research is less clear cut than is presented in the introduction
Line 142: “Past PTSD” – unclear if this means previous history of PTSD or current PTSD lasting <5 or >5 years
Line 145: Recruitment method must be specified – where were participants recruited from
Section 2.1 should describe the recruitment methods, currently it describes results of the collected data
Line 179 – PSQI reference needed
Unclear how addictive and compulsive behaviours were measured
Median split (for age) is not best practice. I don’t know why this analysis is not done instead as a correlation- this is far more valid & median split is rarely accepted as appropriate these days
Why are all participants male? Not clear
Line 279: full stop missing
Why is NMDA receptor in the methods?
Line 326-333 is repetitive
Posthoc tests for Table 2 and onwards are not clearly presented
Line 363-378 is not concise (repetition) and should be cut
There are so many tests run in this manuscript, but some corrections do not seem to be properly applied – eg results of p = .048 should not be significant. There are literally hundreds of tests run so this must be done correctly. Many of the tests in section 3.4 fall into this category
Analyses in section 3.4 are not correctly conducted. See Nieuwenhuis et al 2011. You cannot infer differernces between the groups without apprpirate statistical tests between the groups
Line 598-607 – unclear how this fits in to the discussion
BMI should be included as a covariate when comparing between the groups, especially for biomarkers
There seems to be an author missing from the authorship list
Author Response
Response to Reviewer 2
Manuscript ID: cimb-3870462
Title: “Biomarker-Sleep Correlations in PTSD: Glutamine, Glutathione, Caspase-1, and BDNF Levels Assessed Using the Pittsburgh Sleep Quality Index Addendum”
Authors: Anna Dorota Grzesińska
Dear Reviewer,
I sincerely thank you for your thorough evaluation of my manuscript and for the constructive and insightful comments. I highly appreciate your recognition of the importance and timeliness of the topic, namely the interaction between oxidative stress, neurotrophic factors, and sleep disturbances in patients with PTSD. Your positive assessment of the study’s design—employing multiple biomarkers (glutamine, glutathione, caspase-1, BDNF) and their correlation with sleep dimensions—encourages me that my work contributes valuable new evidence to understanding the biological substrate and chronification of PTSD.
Guided by your suggestions, I have carefully revised the manuscript. Below I provide a detailed, point-by-point response, indicating the modifications made and where they appear in the revised text.
- Reviewer’s comment: The author presents the relationship between biomarkers and PTSD as definitive, but the research is less clear cut than is presented in the introduction
Response: Thank you for this valuable comment. I have revised the Introduction and Discussion, and emphasized the limitations to better reflect the scope of the data.
- Reviewer’s comment: Line 142: “Past PTSD” – unclear if this means previous history of PTSD or current PTSD lasting <5 or >5 years
Response: I have revised the phrasing. The text now provides a clear and unambiguous description of the study groups.
- Reviewer’s comment: Line 145: Recruitment method must be specified – where were participants recruited from
Response: I added a detailed description of recruitment sources and time window, inclusion/exclusion criteria, and screening procedures in Methods, Section 2.1.
- Reviewer’s comment: Section 2.1 should describe the recruitment methods, currently it describes results of the collected data
Response: Corrected. Section 2.1 now focuses on recruitment and eligibility; descriptive data were moved to Results
- Reviewer’s comment: Line 179 – PSQI reference needed
Response: Added the appropriate PSQI-A development/validation references at first mention and in the reference list.
- Reviewer’s comment: Unclear how addictive and compulsive behaviours were measured
Response: I thank the reviewer for pointing out the need for clarity on the measurement of addictive and compulsive behaviors. To address this, I note that these behaviors were assessed through structured clinical interviews based on DSM-5 criteria for impulse-control and non-substance-related addictive disorders, as described in Section 2.1 (Characteristics of the Participants) of the manuscript. The interviews specifically included items evaluating compulsive eating, intrusive trauma-related visualization, excessive exercise, and cyberaddiction, conducted by trained clinical psychiatrists to ensure standardized administration and scoring. This method allowed for a comprehensive evaluation of behavioral patterns relevant to PTSD comorbidity.
- Reviewer’s comment: Median split (for age) is not best practice. I don’t know why this analysis is not done instead as a correlation- this is far more valid & median split is rarely accepted as appropriate these days
- Response: I appreciate the reviewer’s concern regarding the use of age stratification and the potential perception of it as a median split, which I acknowledge is generally discouraged in modern statistical practice due to loss of information and reduced power (e.g., MacCallum et al., 2002). To clarify, the division into age subgroups (18–35 years and 36–50 years) was not derived from a median split of the sample data—although the overall median age (approximately 34 years) was incidentally close—but was instead predetermined based on established developmental and clinical distinctions in PTSD research. This stratification was chosen to explore potential differences in PTSD symptom profiles, biomarker responses, and sleep disturbances between emerging/young adulthood (18–35 years), a period often characterized by higher acute trauma reactivity, ongoing neurodevelopmental maturation, and greater vulnerability to initial PTSD onset, and mid-adulthood (36–50 years), typically associated with accumulated chronic stress, altered coping mechanisms, and potentially more persistent or exacerbated symptoms due to life-stage factors such as occupational demands or cumulative allostatic load.
This approach aligns with prior studies that have identified age-related variations in PTSD prevalence, severity, and neurobiological correlates across the lifespan (e.g., Asselmann et al., 2016; Ditlevsen & Elklit, 2010). Such stratification has been employed in similar investigations to account for these distinctions without relying on data-driven splits (e.g., Bonsaksen et al., 2022, who used identical subgroups in a study of PTSD and visual impairment). As an alternative to subgroup analysis, I considered correlational approaches with age as a continuous variable; however, the stratified design was prioritized to facilitate clinically meaningful interpretations, such as age-specific intervention needs, while preserving power for subgroup comparisons. I have revised the manuscript to explicitly state this rationale and included the relevant references for transparency.
Asselmann E, Stender J, Grabe HJ, König J, Schmidt CO, Hamm AO, Pané-Farré CA. Assessing the interplay of childhood adversities with more recent stressful life events and conditions in predicting panic pathology among adults from the general population. J Affect Disord. 2018 Jan 1;225:715-722. doi: 10.1016/j.jad.2017.08.050
Ditlevsen, D. N., & Elklit, A. (2010). The combined effect of gender and age on post traumatic stress disorder: Do men and women show differences in the lifespan distribution of the disorder? Annals of General Psychiatry, 9, 32. https://doi.org/10.1186/1744-859X-9-32
MacCallum, R. C., Zhang, S., Preacher, K. J., & Rucker, D. D. (2002). On the practice of dichotomization of quantitative variables. Psychological Methods, 7(1), 19–40. https://doi.org/10.1037/1082-989X.7.1.19
Bonsaksen T, Brunes A, Heir T. Post-Traumatic Stress Disorder in People with Visual Impairment Compared with the General Population. Int J Environ Res Public Health. 2022 Jan 6;19(2):619. doi: 10.3390/ijerph19020619. PMID: 35055443; PMCID: PMC8775682.
- Reviewer’s comment: Why are all participants male? Not clear
Response: The cohort was recruited from a male occupational population (details added in Section 2.1), which resulted in an all-male sample. I acknowledge this in Limitations.
- Reviewer’s comment: Line 279: full stop missing
Response: Corrected.
- Reviewer’s comment: Why is NMDA receptor in the methods?
Response: The NMDA-related content was included in error and has been removed.
- Reviewer’s comment: Line 326-333 is repetitive
Response: The passage has been condensed to remove redundancy.
- Reviewer’s comment: Posthoc tests for Table 2 and onwards are not clearly presented
Response: To address this concern, we have revised the Statistical Analysis subsection to explicitly detail the post-hoc procedures for each data type, distinguishing between non-parametric tests (Dunn's test with Bonferroni correction for demographic, addiction-related, and biomarker data) and parametric tests (Games-Howell test with Holm-Bonferroni correction for PSQI-A scores following Welch's ANOVA). Furthermore, we have enhanced the notes accompanying each table to reiterate these methods, provide a clear explanation of the compact letter display (e.g., groups sharing the same letter do not differ significantly at adjusted p > 0.05), and include examples of pairwise comparisons where relevant.
- Reviewer’s comment: Line 363-378 is not concise (repetition) and should be cut
Response: Thank you for pointing this out. The section between lines 363–378 has been revised to remove repetition and is now more concise.
- Reviewer’s comment: There are so many tests run in this manuscript, but some corrections do not seem to be properly applied – eg results of p = .048 should not be significant. There are literally hundreds of tests run so this must be done correctly. Many of the tests in section 3.4 fall into this category
Response: I thank the reviewer for highlighting the issue of multiple testing corrections, which is indeed critical given the number of analyses conducted. To address this, I clarify that corrections for multiple comparisons were selectively applied based on the nature of the analyses: Bonferroni or Holm-Bonferroni adjustments were used for group comparisons (e.g., Kruskal-Wallis post-hoc tests and Welch's ANOVA post-hoc tests) and age-subgroup differences, as detailed in the Statistical Analysis subsection. However, for the correlation analyses in Section 3.4, which were exploratory in purpose and aimed at hypothesis generation rather than confirmatory testing, no such corrections were applied. This approach aligns with established recommendations for exploratory research to avoid overly conservative thresholds that could obscure potential patterns warranting further investigation (Rothman, 1990). The specific example of p = 0.048 refers to a correlation coefficient in this exploratory context and was thus reported without adjustment. I have revised the Statistical Analysis subsection (Section 2.7) to explicitly state this distinction for greater transparency and have incorporated the update into the revised manuscript.
- Reviewer’s comment: Analyses in section 3.4 are not correctly conducted. See Nieuwenhuis et al 2011. You cannot infer differernces between the groups without apprpirate statistical tests between the groups
Response: I appreciate the reviewer’s reference to Nieuwenhuis et al. (2011) and their concern regarding the inference of group differences in correlations without formal statistical testing. I agree that subsection 3.4.4 ("Major differences in correlations among groups") primarily involves descriptive comparisons of correlation patterns across groups, rather than inferential tests of differences between those correlations. No direct statistical comparisons (e.g., Fisher's z-transformation or similar methods) were performed to assess whether correlation coefficients differed significantly between groups, as the analysis was intended to be exploratory. The goal was to highlight observed patterns, such as the presence or absence of statistical significance in one group versus others, or relatively stronger associations within groups, to generate hypotheses for future research. To address this, I have revised the subsection title to "Patterns of correlations across groups" for clarity and tempered the language to emphasize its descriptive and exploratory nature, avoiding implications of tested differences. Phrases suggesting progression, decoupling, or shifts have been rephrased to focus on observed variations without inferring causality or statistical contrasts.
- Reviewer’s comment: Line 598-607 – unclear how this fits in to the discussion
Response: The paragraph was reworked and relocated to the most appropriate subsection; content not essential to our aims was removed. Cross-references were updated for clarity.
- Reviewer’s comment: BMI should be included as a covariate when comparing between the groups, especially for biomarkers
Response: Response: I appreciate the reviewer’s recommendation to incorporate BMI as a covariate in group comparisons, particularly for biomarkers, to account for potential confounding effects on physiological outcomes. To address this, I have revised the statistical approach and re-estimated the results for Tables 2 and 3. Specifically, for biomarker data in Table 2 (glutamine, glutathione, caspase-1, and BDNF), I applied Quade's rank-based analysis of covariance (ANCOVA), a non-parametric method that adjusts for BMI while accommodating non-normality. Post-hoc pairwise comparisons were conducted using Quade's test on group subsets, with Holm-Bonferroni correction for multiple comparisons. For PSQI-A scores in Table 3, I employed ANCOVA to adjust for BMI, using Welch's ANOVA to handle unequal variances, followed by Games-Howell post-hoc tests with Holm-Bonferroni correction. These updates have been integrated into the revised manuscript, including the Statistical Analysis section for transparency. The re-estimated results largely confirm the original patterns of group differences, with minor adjustments in p-values but no substantive changes to the overall conclusions regarding PTSD chronicity and biomarker or sleep profiles. I will include the updated tables in the resubmission.
- Reviewer’s comment: There seems to be an author missing from the authorship list
Response: Thank you for your comment. I confirm that there is only one author of this article.
I believe that the revisions made significantly strengthen my manuscript, improving both its clarity and scientific value. I thank you once again for your insightful review and constructive suggestions, which have allowed me to enhance the quality of my work.
Sincerely,
Anna Dorota Grzesińska

Round 2
Reviewer 1 Report
Comments and Suggestions for Authors
The authors successfully answered all my concerns.
Author Response
Dear Reviewer,
Thank you very much for your valuable feedback and constructive comments, which have helped me to improve the manuscript. I truly appreciate the time and effort you dedicated to the review.
With kind regards,
Anna Grzesińska
Reviewer 2 Report
Comments and Suggestions for Authors
Thanks for the revision. I think that this generally meets the review that I suggested. I would say that the author should additionally emphasise the exploratory nature of the analyses. I also do not understand why posthoc corrections for multiple comparisons would not apply to exploratory tests. A citation for this decision should be provided as it leverages some of the results of the tests. Otherwise I am happy to accept
Author Response
CIMB (ISSN 1467-3045)
cimb-3870462
Thank You for your valuable feedback and for Your overall positive evaluation of my manuscript. I appreciate Your suggestion to further emphasize the exploratory nature of the analyses and to provide a clearer justification for not applying post-hoc corrections for multiple comparisons in the correlation analyses. These points strengthen the statistical rigor and transparency of my work, particularly in the context of exploratory biomarker-sleep correlations in PTSD, where hypothesis generation is prioritized over confirmatory testing.
To address Your concerns, I have revised the manuscript as follows:
- I have expanded the emphasis on the exploratory nature of the analyses in Section 2.7 (Statistical Analysis) and Section 4 (Discussion) to clearly frame the correlations as preliminary and intended for generating insights into potential pathophysiological mechanisms, rather than establishing definitive associations.
- Regarding the lack of post-hoc corrections for the correlation analyses, I have enhanced the justification in Section 2.7. This decision aligns with established statistical guidance for exploratory research, where such adjustments can inflate Type II error rates and obscure potentially meaningful patterns in novel areas like PTSD biomarker profiles. I have retained and elaborated on the citation to Rothman (1990), which supports omitting adjustments in hypothesis-generating contexts to maintain sensitivity, especially with modest sample sizes and interdependent variables (e.g., biologically linked biomarkers). From a clinical perspective, this approach allows identification of subtle trends that could inform future targeted interventions in PTSD, such as addressing oxidative and inflammatory pathways linked to sleep disturbances, while acknowledging the need for confirmatory studies to validate findings.
These revisions are detailed below and incorporated in the tracked changes version of the manuscript. I believe they fully resolve your points from both statistical and clinical viewpoints, ensuring cautious interpretation without over-correction that might hinder discovery.
I am grateful for Your time and insights, which have improved the manuscript.
Anna Grzesińska
